# Effect of Forming and Heat Treatment Parameters on the Mechanical Properties of Medium Manganese Steel with 5% Mn

**DOI:** 10.3390/ma16124340

**Published:** 2023-06-12

**Authors:** Radek Leták, Hana Jirková, Ludmila Kučerová, Štěpán Jeníček, Josef Volák

**Affiliations:** 1Regional Technological Institute, University of West Bohemia, Univerzitní 8, 301 00 Pilsen, Czech Republic; skal@fst.zcu.cz (L.K.); jeniceks@fst.zcu.cz (Š.J.);; 2Department of Materials and Engineering Metallurgy, University of West Bohemia, Univerzitní 8, 301 00 Pilsen, Czech Republic; hstankov@fst.zcu.cz

**Keywords:** medium manganese steels, press hardening, induction heating, omega profiles

## Abstract

Medium manganese steels fall into the category of modern third-generation high-strength steels. Thanks to their alloying, they use a number of strengthening mechanisms, such as the TRIP and TWIP effects, to achieve their mechanical properties. The excellent combination of strength and ductility also makes them suitable for safety components in car shells, such as side reinforcements. Medium manganese steel with 0.2% C, 5% Mn, and 3% Al was used for the experimental program. Sheets with a thickness of 1.8 mm without surface treatment were formed in a press hardening tool. Side reinforcements require various mechanical properties in different parts. The change in mechanical properties was tested on the produced profiles. The changes in the tested regions were produced by local heating to an intercritical region. These results were compared with classically annealed specimens in a furnace. In the case of tool hardening, strength limits were over 1450 MPa with a ductility of about 15%.

## 1. Introduction

The automotive industry is one of the largest consumers of pressed parts. This industry has an annual demand for pressed products of around 58% of all world production. This figure increases by leaps and bounds every year [1]. The automotive industry is pushing forward the development of advanced high-strength steels (AHSS) that features good strength, ductility, and formability [2,3,4]. Today’s trend in the automotive industry is to reduce the weight of individual parts of the car body while maintaining one of its main characteristics, namely passenger safety. The development teams of the world’s automotive companies are therefore looking for ways to use new materials to reduce the weight of the entire body, leading to lower fuel consumption and thus reducing the carbon footprint and meeting the EU’s green values [5,6,7,8,9]. It is important to combine the weight reduction in the bodywork without losing its ability to protect the occupants during an impact. A lot of teams are working on this problem [10,11]. In addition to lower weight, emphasis is also placed on design, high production efficiency, and ease of assembly. It must be possible to carry out technological surface treatments on the material. For this reason, it is necessary to exclude certain elements that would make this impossible [12]. Unfortunately, all these requirements must be met under very strict economic conditions. One of the first and main problems is the combination of good formability with high strength, which are two opposing properties to be achieved by heat treatment [8].

In addition, some body parts, such as the B-pillars, need different mechanical properties in different locations. If the components have different properties, they are more valuable for the safety of the car in a crash. Automotive B-pillars, as seen in Figure 1, must exhibit a gradient strength distribution [13,14]. This is characterized by different properties at the top and the bottom of the product. The upper part has high strength and lower ductility, so the body does not deform sharply in the event of a collision. This ensures a safe space for passengers. The lower part with lower strength and higher ductility absorbs the energy generated in a collision. The production of such tailored parts is very complex and uses various methods [10,15,16,17].

The production of complex sheet metal parts led to the intensive development of press hardening technology. With this technology, it is possible to process ultra-high strength hardenable sheets that are developed for heat treatment. This type of steel contains 22MnB5-based steels, which have been developed especially for this technology. These are manganese boron steels which, due to their composition, achieve a martensitic structure and a strength limit of around 1500 MPa during press hardening [15,19,20,21].

Modern high-strength steels are also increasingly being used. Currently, a group of steels is being developed, the so-called third generation, which includes medium manganese steels (MMnS), characterised by a manganese content of between 3 and 12%. Medium manganese steels are characterised by improved mechanical properties compared to first-generation steels and at the same time offer cost savings by reducing the content of alloying elements and increasing productivity compared to second-generation steels [8,22].

For these steels, the achievement of the desired properties is strongly influenced by the heat treatment process. The key technologies include intercritical annealing (IA), where the annealing temperature is set between A_c1_ and A_c3_, quenching and partitioning (QP process), martensite to austenite reverse treatment, quenching–partitioning–tempering (Q–P–T), and dual stabilisation heat treatment (DSHT). The mechanical properties are strongly influenced by the proportion of stabilised retained austenite, which is affected by the IA temperature and the holding time at this temperature. It has been shown that IA annealing does not lead to a homogeneous distribution of manganese in the austenitic phase. The amount of manganese in austenitic grains varies by up to 5% after IA annealing. This difference in chemical composition is due to the formation of austenitic grains and the deposition of manganese. Additionally, this effect promotes the stability of RA [6].

The mechanical properties correspond directly to the characteristics of the residual austenite. Characteristics, such as the amount and morphology of residual austenite in MMnS steels, depend on the annealing process. At low annealing temperatures, the residual austenite shows good thermal stability, which is related to a large amount of stabilising manganese. By contrast, the kinetics of the reverse conversion from martensite to austenite is slow at low temperatures. When the annealing temperature is increased, the kinetics of the conversion of atomic diffusion of manganese and carbon becomes faster [1,6].

Intercritical annealing (IA), taken as an effective way to avoid strength-ductility, discontinuities in this alloy system. Introducing metastable retained austenite (γ_R_) into the two-phase microstructure of ferrite (α) and martensite (α′) thereby generates a strain-induced plasticity (TRIP) effect that delays localized necking and prevents damage initiation [6]. The steel types are shown in Figure 2 [5].

In the conventional press hardening process, samples are heated in a furnace above the primary conversion temperature of ferrite to austenite (A_c3_) with a temperature endurance typically in the 890–950 °C range [23,24,25,26]. The processed materials are then transferred to a press equipped with a tool, which in most cases is water-cooled. Here, forming and hardening take place simultaneously at supercritical cooling rates of between 40 and 25 K/s [18]. This thermomechanical process results in a fully martensitic structure with a tensile strength of up to 1500 MPa [8]. In hot pressing, lower forming forces have to be used compared to cold forming, the material has a higher ductility, and the spring back effect is also very small, around 5% [11].

The production of moulded parts by press hardening with tailor-made properties, so-called tailoring has become an area of research interest in the automotive industry. Current methods for obtaining hot press parts with different properties fall into several categories. The first method is the joining of two dissimilar materials by laser welding. This produces original sheet metal parts with different strengths or thicknesses. Another way is to change the heating or cooling parameters during the hot-pressing process. In this case, one part of the hardening tool is heated and another part of the forming tool is cooled [15,16,18]. One other possibility is the inclusion of a device allowing local annealing. Such a device is used after the final shape has been formed in the hot stamping tool. With these methods, parts with different properties can be obtained. These are obtained by precise control of the microstructure during cooling and thus have significant application prospects.

As mentioned above, there are many techniques to obtain different structures and thus mechanical properties on a single part [27,28]. The question is which procedure will lead to the best combination of the desired properties. Given the requirements for a wide variance in mechanical properties, newly developed third-generation high-strength steels, such as medium manganese steels, are suitable for these parts. The use of these steels for this type of part raises many questions and areas of research due to the complexity of the heat treatment procedure and the need to obtain the necessary proportion of residual austenite to ensure the required mechanical properties.

Therefore, this paper deals with the comparison of different heat treatment procedures after forming in a press hardening tool and describes the different possibilities to obtain tailored properties.

## 2. Materials and Methods

The medium manganese steel 2C5Mn3Al was used for the experimental program (Table 1). This steel contained 0.2% C and other alloying elements, including main manganese with 5% and aluminium with almost 3%. Manganese together with carbon stabilizes the austenite. Aluminium contributes to faster reverse austenitic conversion. Silicon reduces the rate of carbide precipitation in bainite, thereby stabilizing the residual austenite. Concerning the chemical composition, this steel has the potential to form a functional part with a multiphase structure containing residual austenite in the matrix. The material used is classified as a so-called third-generation AHSS steel. For the final product, the desired mechanical properties are achieved by a combination of TRIP and TWIP effects, the latter depending on the magnitude of the stacking fault energy (SFE). The chemical composition is given in Table 1.

The experimental steel was cast into a conical cone-shaped ingot weighing 168 kg. Afterwards, the surface was machined to remove the scale and a sheet thickness of 4.2 mm was obtained. The final cold rolling was carried out in 16 steps with a reduction size of 5%. The final sheet thickness was 1.8 mm. From the sheets prepared this way, samples of 120 × 80 mm were cut by waterjet. The final step was a homogenization annealing in a protective argon atmosphere at 680 °C with a residence time of 2 h.

The phase transformation temperatures were determined by calculation in JMatPro software and are shown in Figure 3. It was found that the temperature of A_c1_ is around 1071 °C, and the temperature of A_C3_ has a value of around 683 °C. The perlitic transformation is significantly shifted towards slower cooling rates (Table 2). The same process, shifting to lower cooling rates, also occurs for the bainitic alteration. The M_s_ temperature was also evaluated from JMatPro. The martensite start temperature was calculated to be 286 °C.

### 2.1. Press Hardening

Press hardening was performed in a tool with a CKW6000 hydraulic press (ŽDAS, a.s., Ždár and Sázavou, Czech Republic). The tool had an omega profile shape with a profile depth of 30 mm (Figure 4 and Figure 5). The processing consisted of heating the sheet metal blank in an electric furnace without a protective atmosphere to a temperature of 1050 °C and a holding time of 30 min. Then, the experimental steel sheet was transferred to a press hardening tool at an RT. The transfer from the furnace to the prepared tool took 2 s. This was followed by closing the instrument. Closing the tool resulted in the creation of an omega profile. After sealing the tool, a tool dwell of 1 or 5 s was performed.

### 2.2. Intercritical Annealing of Omega Profiles

Enhancement of the ductility value can be achieved by a suitable heat treatment process. Therefore, the omega profiles prepared in the previous press hardening step with a tool holding time of 5 s were subjected to intercritical annealing.

The intercritical temperatures were chosen at 50 °C intervals: 700, 750, and 800 °C. The lowest temperature of 700 °C was close to the A_c1_ curve. The holding time was 30 min. All selected temperatures were chosen in the intercritical region, i.e., between the A_c1_ and A_c3_ temperatures. These values were read from the CCT diagram for the experimental steel in Figure 3.

### 2.3. Production of an Omega Profile with Locally Different Properties

The next step was to try to achieve different characteristics within a single part. The forming in the tool was performed by a combination of pressing and free cooling in the air. After heating the sheet metal part to a temperature of 1050 °C, the specimen was inserted into the room-temperature, press hardening tool with only one-half of the part. In this half of the omega profile, both forming and intensive cooling due to heat dissipation into the cold tool were performed (Figure 6). The other half of the sheet was placed outside the tool and cooled freely in the air. After a specified time, the tool was opened, and the entire blank was transferred into the tool to form the entire omega profile.

After the first processing step, the part of the sheet, which was formed and quenched in the tool, had its temperature already below the M_s_ temperature. The temperature of the sheet part cooled as the air dropped into the intercritical region, which corresponds to temperatures between A_c3_ and A_c1_. The air-cooling times were chosen according to the width of the temperature interval for the intercritical region, so that in the air-cooled part, partial ferrite formation had already transformed before it was transferred to the tool and closed. A K-type thermocouple was used to measure the temperatures and was welded to the surface of the sheet. The thermocouples were used for the part cooled first at the air. At the end of the processing, the entire omega profile was imaged with a thermal imaging camera to determine the temperature distribution in the individual parts.

Using a thermocouple, the cooling curve of the sheet in the air was determined (Figure 7). Subsequently, the intercritical temperatures were determined from the CCT diagram. From the intersections of these cooling curve temperatures, the quench times in the tool were determined. These times were 5 s, 10 s, and 300 s. After transferring the entire sheet into the tool, quenching was performed for 5 s. This time was the same for all samples.

These special omega profiles were sampled for analysis from three different locations. The first area examined was the area first cooled in air, the second area examined was the area straight hardened in the tool, and the third area was the transition area, between the previous two (Figure 8).

### 2.4. Induction Heating

In the last stage of the experiment, the mechanical properties were changed only locally in a predefined region. The input for this part of the experiment was again the omega profile obtained by quenching in the tool for 5 s. Local annealing was performed after cooling the omega profile at RT by induction heating. This was a DHI 100 F induction heater from Dawell, which has a maximum power of 8 kW and an operating frequency from 18 to 45 kHz. The induction power was used at 25% with a residence time of 6 s. The temperatures achieved were in the range of 870 to 930 °C. The local heating was designed to simulate intercritical annealing in a local area and to significantly increase the ductility value in that area. The temperature was measured using a K-type thermocouple as well as a thermal camera, from which the images shown in Figure 9 are taken.

### 2.5. Evaluation Methods Applied

In total, 1 omega profile with a tool holding time of 1 s and 4 omega profiles with a tool holding time of 5 s were produced. Three of these profiles were then used for other parts of the experiment (intercritical annealing and local induction heating). Metallographic samples in the transverse direction were prepared from each omega profile, and hardness measurements and tensile tests were performed. For the tensile test, we used two samples. Furthermore, three omega profiles were produced for the “production of an omega profile with locally different properties” part, where the endurance parameters were selected.

The evaluation of the microstructures of the experimental parts after each stage was carried out on transverse sections using a light microscope Olympus (Olympus, Tokyo, Japan). A Tescan scanning electron microscope was used for detailed microstructural analysis (Tescan, Brno, Czech Republic). The samples were prepared by standard metallographic methods and the structure was highlighted using Vilella-Bain etching. The hardness of HV10 was also measured on metallographic cuttings. The values shown are averages of five measured values. Vickers hardness measurements (LECO Instrumente Plzeň, spol. s r. o., Pilsen, Czech Republic) were performed with a load of 10 kg.

The tensile test was performed according to Č SN EN ISO 6892-1 method A on a ZWICK 250 universal machine with a maximum force of 250 KN (ZWICK 250, Ulm, Germany). Two samples were tested from each parameter variation. Samples were cut in the longitudinal direction (Figure 8). The geometry of the bodies for the mini-tensile test is shown in Figure 10. The length of the active part was 5 mm, and the cross section was 2 × 1.5 mm.

## 3. Results

The metallography of the initial states was evaluated after cold rolling and annealing at 680 °C/1 h. The structure consisted of ferritic matrix and spheroidized carbides (Figure 11). The hardness value was 227 ± 7 HV10.

### 3.1. Press Hardening

Press hardening was the initial step for all subsequent operations. The specimen was hardened in the tool for 1 or 5 s. In the case of a short holding time, the structure consisted of a mixture of ferrite, martensite, and a small amount of bainite with a hardness of 415 HV10 (Figure 12). The ultimate strength and ductility were 1362 MPa and 17.7%, respectively (Table 3). When the quenching time in the tool was extended to 5 s the structure was martensitic–ferritic, which was reflected in an increase in the hardness value to 431 HV10. The ultimate strength reached 1404 MPa with a ductility of 16% (Table 3). Therefore, tool hardening with a tool life of 5 s was selected as the basic procedure for the production of omega profiles for the following heat treatment steps.

### 3.2. Intercritical Annealing

The aim of this process was to increase the ductility of the material and to obtain a multiphase structure consisting of a mixture of ferrite, martensite, and residual austenite. After quenching, the samples were successively annealed at 700, 750, and 800 °C (Table 4). In the structures of the samples annealed at 700 and 750 °C, a bainitic-type structure was formed, and retained austenite was observed among the needles formed. The martensitic regions were also significantly higher. The hardness value dropped from the original 433 HV10 after quenching in the tool to 273 HV10 at annealing temperatures of 700 °C and 335 HV10 at annealing temperatures of 750 °C, respectively (Figure 13). The structure of the sample annealed at 800 °C contained a higher fraction of martensite, and the proportion of bainite was significantly lower here, which is also due to the higher hardness value of 377 HV10 (Figure 13). The higher heating temperature led to the formation of a higher fraction of austenite in the structure, the amount of which could not be sufficiently stabilised by the carbon present in the steel, and therefore, austenite mainly transformed back into martensite during the cooling. The higher fraction of martensite in the structure led to the highest ultimate strength of 1425 MPa (Table 4).

### 3.3. Tailoring

For the profile that was processed by a combined procedure with a 5 s cooling outside the tool followed by a 5 s cooling in the tool, the same structure was found in all areas. It was a structure with the presence of ferritic and austenitic grains (Figure 14). The lowest hardness value of 480 HV10 was measured for the part of the omega profile that was first cooled for 5 s in air and only then formed in the tool. On the contrary, the part of the omega profile that was cooled in the tool from the beginning showed the highest hardness of 506 HV10 (Figure 14). It can be assumed that a partial transformation of austenite to martensite occurred in the spherical austenite grains.

Extending the residence time in the air to 10 s did not lead to a significant change in structure in the case of the combination of the 10 s and 5 s treatments. The structure in the part cooled in the air for 10 s was again composed of a martensitic matrix with a proportion of ferritic and austenitic grains (Figure 14 and Figure 15). In the austenitic grains, a significant transformation to martensite was already evident, and these grains were thus composed of the M–A component. The hardness value in this region was 477 HV10 (Figure 15). In the part of the omega profile that was cooled directly in the tool, needles of secondary martensite were again detected in the original austenitic islands. Cooling directly from the austenitic region resulted in a higher proportion of martensite in the structure, which was reflected in a higher hardness value of 528 HV10 (Figure 15). In the case of the 30 + 5 s treatment combination, very similar values of structure and hardness were achieved for all parts examined (Figure 16).

The highest strength limits were found in the areas cooled directly in the tool throughout the treatment. In the case of 5 + 5 and 10 + 5 processing, the strength limit was higher than 1630 MPa at a ductility of approximately 15% (Figure 15). In the case of 30 s air cooling time, the strength limit increased to 1575 MPa (Table 5), which was due to the conversion of more austenite to martensite. Cooling was interrupted at 695 °C for the 5 + 5 s treatment and at 645 °C for the 10+5 treatment, while cooling was interrupted at 520 °C for the 30 + 5 treatment.

### 3.4. Induction Heating

The sample was quenched in the tool for 5 s and was locally heat-treated by induction heating. The sample was locally annealed with a flat coil for 6 s. Subsequent cooling was carried out in the air. The annealing temperature was found to be 925 °C using a thermal imaging camera and a thermocouple (Figure 9). From the induction annealing area, not only was a sample taken for metallographic analysis but also two mini-plates for mechanical testing. The structure at the induction heating site consisted of a mixture of martensite and ferrite with a small amount of residual austenite (Figure 17). Compared to the original state after press hardening, the proportion of ferrite in the structure increased. In terms of mechanical properties, the ultimate strength decreased from 1440 MPa to 1219 MPa (Table 6).

## 4. Discussion

After press hardening, the aim is to achieve a mainly martensitic structure with an ultimate strength of around 1500 MPa, as is usual when using 22MnB5-based steels, which were developed especially for this technology [30]. The problem with these steels is the value of ductility, which varies after hardening in the tool below 5%.

In the proposed medium manganese steel, a high ultimate strength value was achieved by increasing the tool life. By selecting a suitable tool hardening time, it is possible to achieve an ultimate strength of 1440 MPa at a ductility of 16%. The higher ductility value of the medium manganese steels is due to the multiphase structure of these steels, which consists of a martensitic matrix with a small proportion of ferrite and residual austenite, which is mostly found along the boundaries of the martensitic needles [1] (Figure 18). The occurrence of residual austenite along the boundaries of martensitic needles in this type of steel was also confirmed in the study [31]. Globular austenite also occurs in this type of steel, which is confirmed by studies [32,33].

The ultimate strength value before the press hardening process is around 550 MPa for 22MnB5 materials [34]. The structure of the material influenced the hardness, which increased with the percentage of martensite in the structure.

The tailoring of a component is used to obtain the desired mechanical properties in definite areas of the part. Especially with B-pillars, it is necessary to increase the value of the ductility in certain areas, for example, to absorb the energy during the impact and avoid risking the passenger’s safety [35]. If a preheated tool is used, for example, it is possible to achieve an increase in ductility of up to 23% for 22MnB5 steel [34]. In the combined processing where part of the sheet metal sample was cooled in air and part was cooled in the tool, a difference in mechanical properties was achieved. This is due to the different microstructure obtained by intermittent quenching [15].

After tailoring, the highest strength limits were measured on a specimen that was processed with a combination of 10 + 5 s. A strength limit of 1640 MPa was reached in the part cooled in the tool for the whole time. The structure was composed of fine martensite and retained austenite. Austenite was detected in the form of thin films along the boundaries of martensitic needles (Figure 19). A similar character of the structure was observed in the case of the work of Giulio Ventura [30]. The decrease in ultimate strength in the region of the sample that was initially cooled in the air was due to the formation of ferritic grains (Figure 20). The ultimate strength dropped to 1440 MPa.

The 30 + 5 specimen appears to be specific, where the ultimate strength value was very similar in all areas studied and was around 1555 MPa. Jing Zhou et al. experimented with different austenitization temperatures and subsequently investigated the surface hardness. Our hardness values were similar to those of Jing Zhou [31].

The results of intercritical annealing in terms of mechanical properties show that the strength limit increases with an increasing heating temperature. At an intercritical annealing temperature of 700 °C, a strength limit of 1261 MPa was reached, and at an IA temperature of 800 °C, the strength increased to 1425 MPa. Heating to the intercritical region resulted in the formation of austenite in the structure. This austenite was not stable, and therefore, part of it transformed to martensite during cooling to RT, resulting in the formation of fresh or secondary martensite. The remaining martensite before IA was heavily tempered, and therefore, a decrease in strength occurred at IA temperatures of 700 and 750 °C compared to the state after press hardening. At IA 800 °C, a large fraction of austenite was already formed, which then transformed into secondary martensite, thus giving the highest ultimate strength value and low values of ductility. The hardness values correspond to the formed structure, with the lowest value of 273 HV10 measured at an annealing temperature of 700 °C and the highest value of 377 HV10 at an annealing temperature of 800 °C. Figure 21 shows the course of the tensile test. The difference between the annealed material and the material after the tailoring process is obvious. The deformation behaviour of the material after the tailoring process is significantly better. The ductility of the material after annealing is only around 6%, but the value for the other processing is slightly above 16%.

Figure 22 shows two fracture surfaces after tensile testing. The left part was a post-processed specimen where part of the specimen cooled for 5 s in air and another 5 s inside the tool. The right part of the image is the sample after intercritical annealing at 750 °C. The samples have different types of fractures when the specimen after tailoring was tougher than the specimen processed by intercritical annealing. This was confirmed by tensile testing.

Induction heating had lower mechanical property values compared to intercritical annealing of the whole sample. Compared to this procedure, a higher heating temperature of 925 °C was achieved by local induction heating. This temperature still lies below the A_c3_ temperature and is therefore still in the intercritical region. In the structure, however, too much austenite was probably already formed during heating, which could not be sufficiently stabilised by manganese and carbon, and during the subsequent cooling in air, it was transformed into ferrite and secondary martensite (Figure 23). It is also evident that the primary martensite was tempered. Therefore, a strength of 1219 MPa was obtained, which is approximately 200 MPa lower than the intercritical annealing temperature of 800 °C of the whole sample. Due to the higher ferrite content and the tempering of the primary martensite, the ductility increased to 14%.

## 5. Conclusions

The medium manganese steel 2C5Mn3Al was processed by press hardening technology, and then the possibilities of modifying the mechanical properties by different processing procedures were tested.

After press hardening, a predominantly martensitic structure with an ultimate strength of 1400 MPa and a ductility of 17% was obtained at longer tool life. Intercritical annealing with heating between A_c1_ and A_c3_ temperatures did not lead to the desired change in mechanical properties, and on the contrary, a deterioration in ductility occurred.

By varying the cooling rate in different parts of the part, tailoring the mechanical properties during the formation of the omega profile was achieved. With a combined treatment of 10 + 10 s, a difference in intergranular strength of up to 200 MPa was achieved, with intergranular strengths of up to 1640 MPa in the part hardened in the tool for the entire time.

## Figures and Tables

**Figure 1 materials-16-04340-f001:**
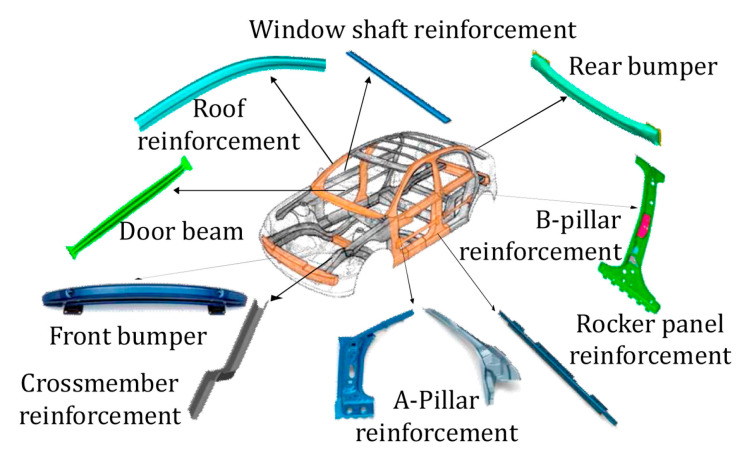
Hot-pressed automotive parts [18].

**Figure 2 materials-16-04340-f002:**
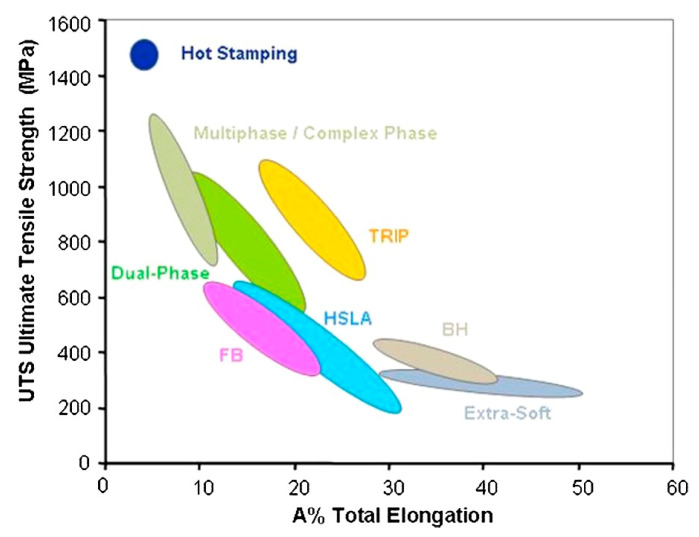
Representation of the main low-alloy steel grades in the tensile strength vs. total elongation diagram. Two-phase, TRIP, and complex phase are considered the first generation of high-strength steels [5].

**Figure 3 materials-16-04340-f003:**
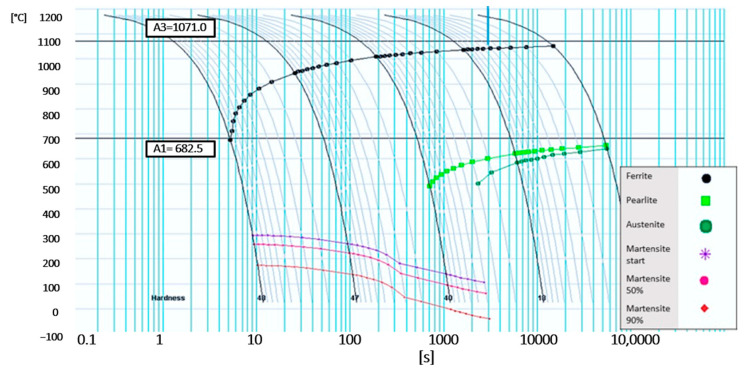
CCT Diagram material 2C5Mn3Al.

**Figure 4 materials-16-04340-f004:**
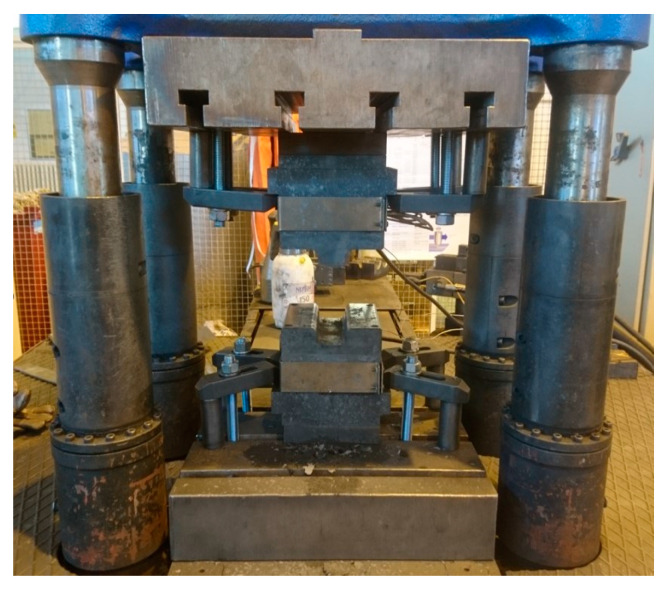
Hydraulic press with tool mounted.

**Figure 5 materials-16-04340-f005:**
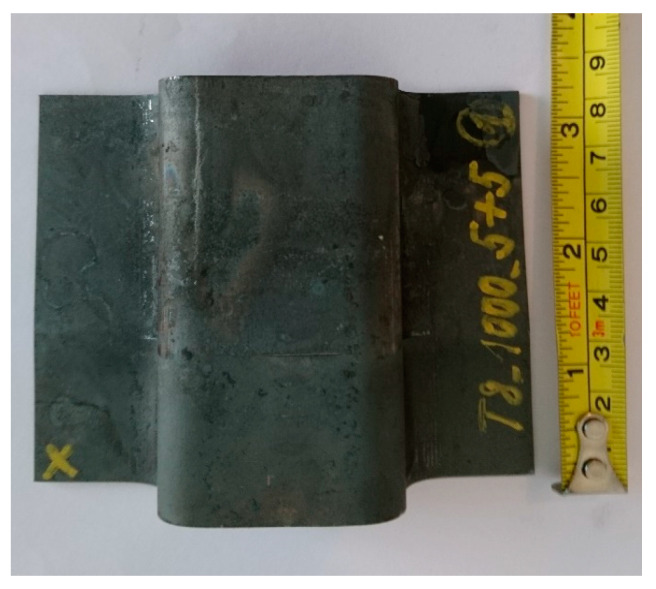
Omega profile.

**Figure 6 materials-16-04340-f006:**
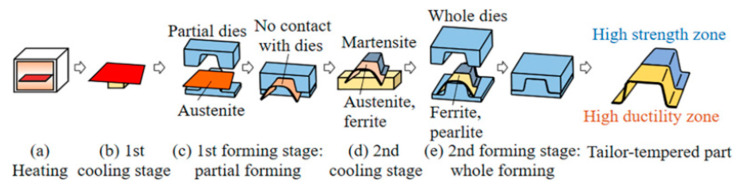
The step-by-step procedure of the tailored process [29].

**Figure 7 materials-16-04340-f007:**
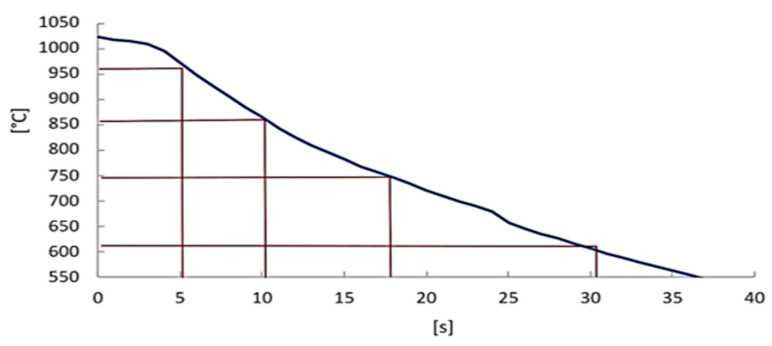
Cooling curve for 2C5Mn3Al during air cooling.

**Figure 8 materials-16-04340-f008:**
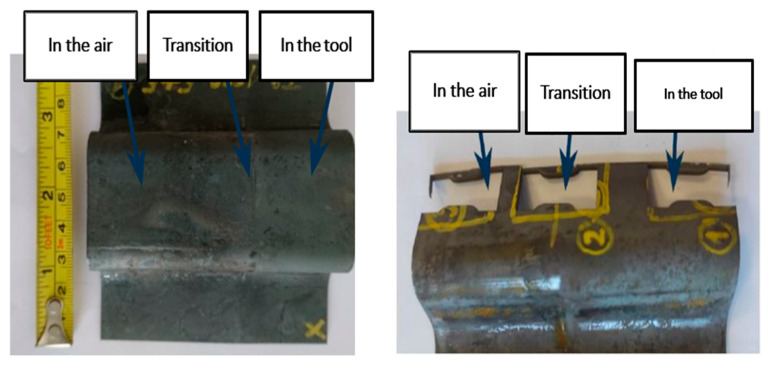
Omega profile with areas sampled for examination.

**Figure 9 materials-16-04340-f009:**
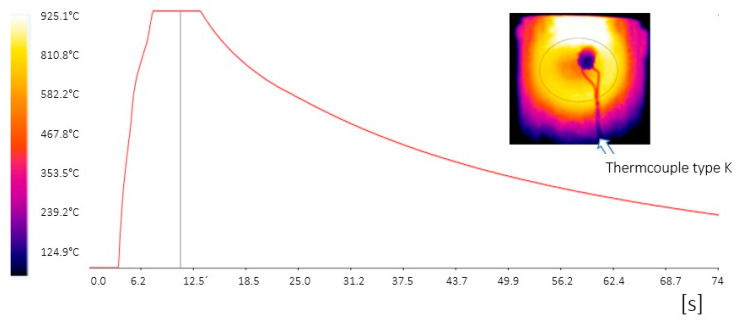
Induction annealing temperature waveform and thermal camera recording.

**Figure 10 materials-16-04340-f010:**
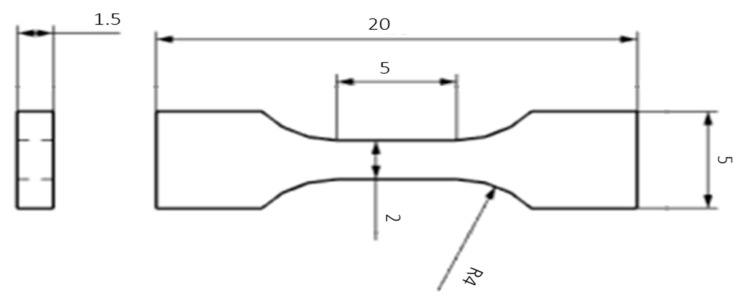
Drawing of the specimen for the mini-tensile test.Unit are in [mm].

**Figure 11 materials-16-04340-f011:**
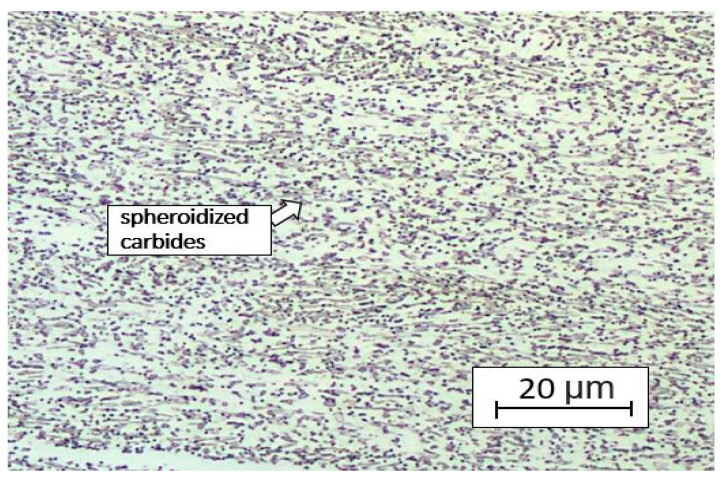
Microstructure of the state after cold rolling and annealing.

**Figure 12 materials-16-04340-f012:**
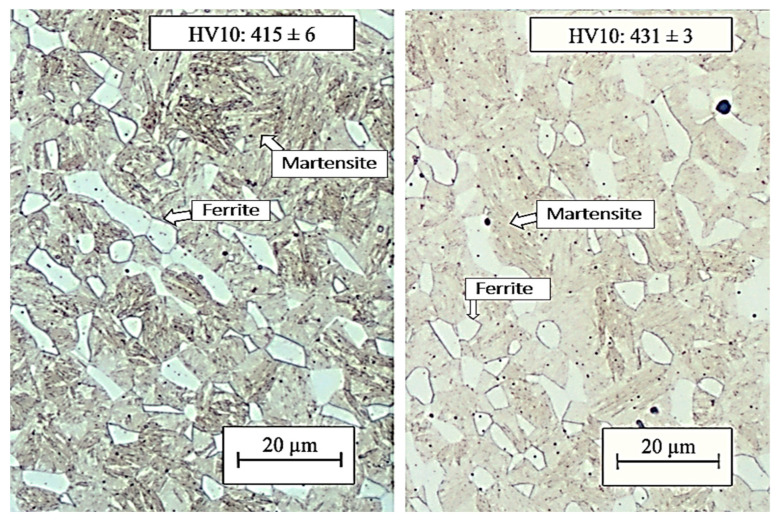
In the left part of the picture is metallography after 1s quenching, and in the right part of the picture is metallography after 5s quenching.

**Figure 13 materials-16-04340-f013:**
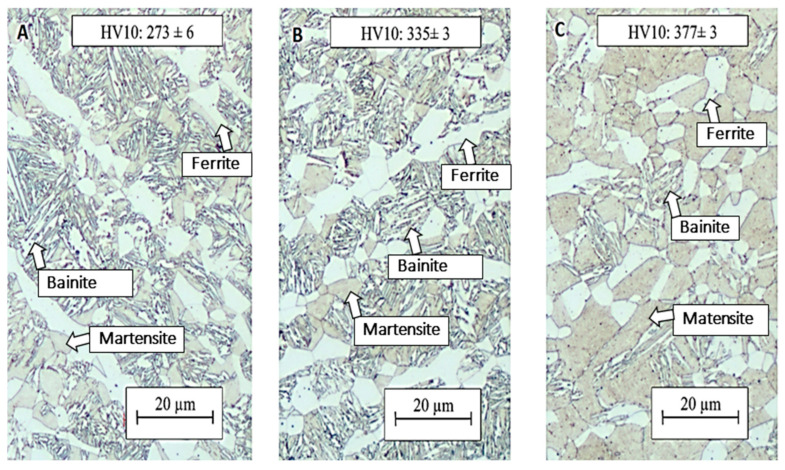
(**A**) Annealed at 700 °C, (**B**) annealed at 750 °C, and (**C**) annealed at 800 °C.

**Figure 14 materials-16-04340-f014:**
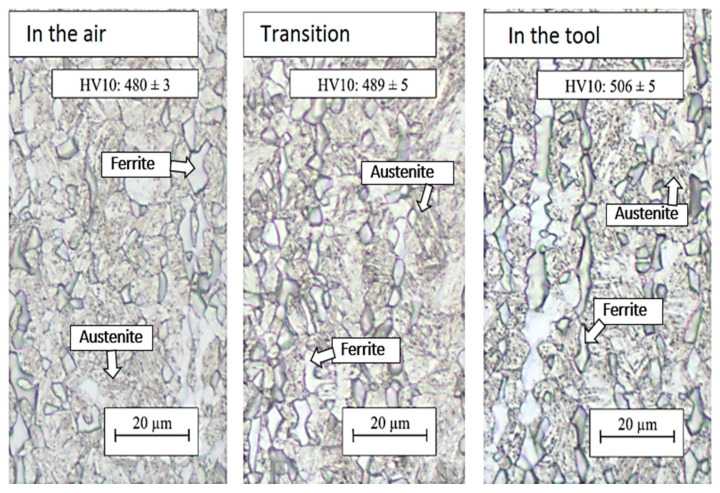
Structures after regime: 5 + 5 s, optical microscopy.

**Figure 15 materials-16-04340-f015:**
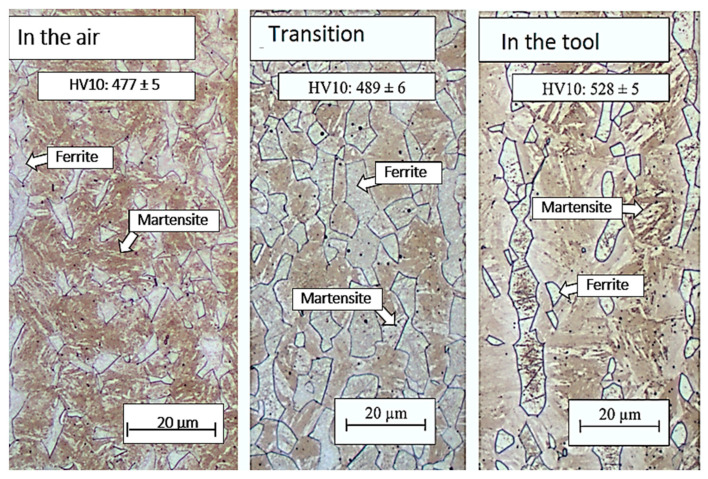
Structures after regime: 10 + 5 s, optical microscopy.

**Figure 16 materials-16-04340-f016:**
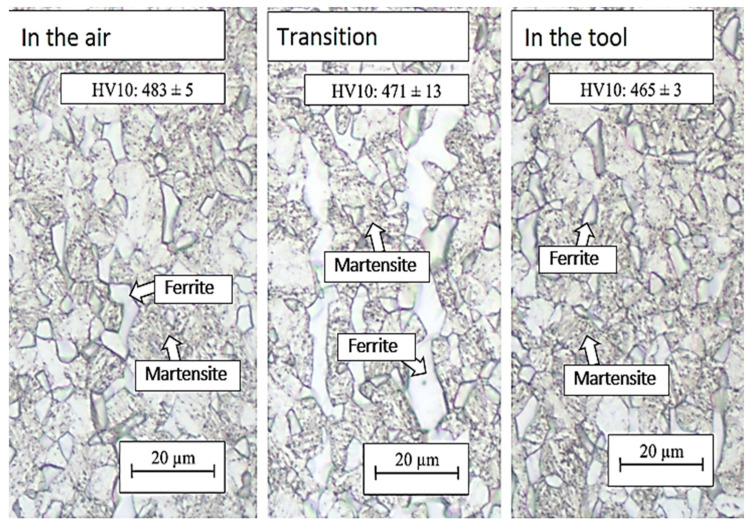
Structures after regime: 30 + 5 s, optical microscopy.

**Figure 17 materials-16-04340-f017:**
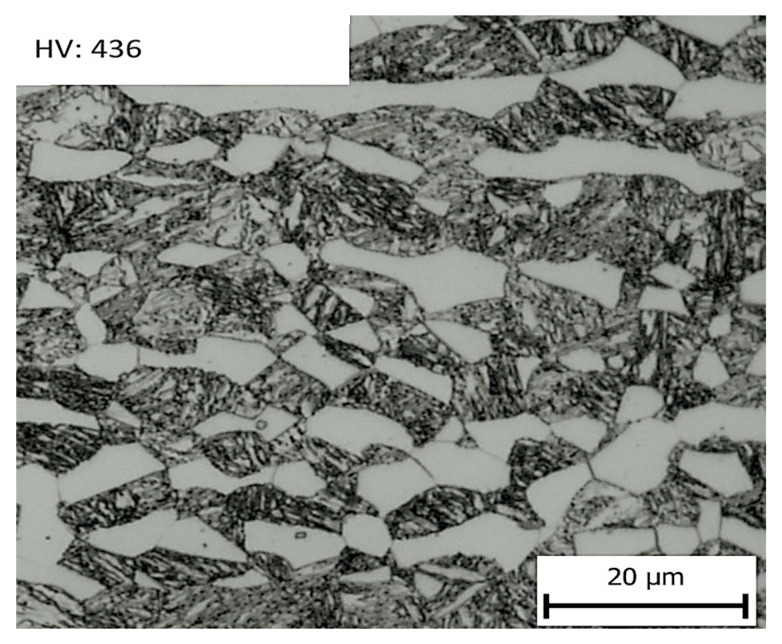
Microstructure after local induction heating.

**Figure 18 materials-16-04340-f018:**
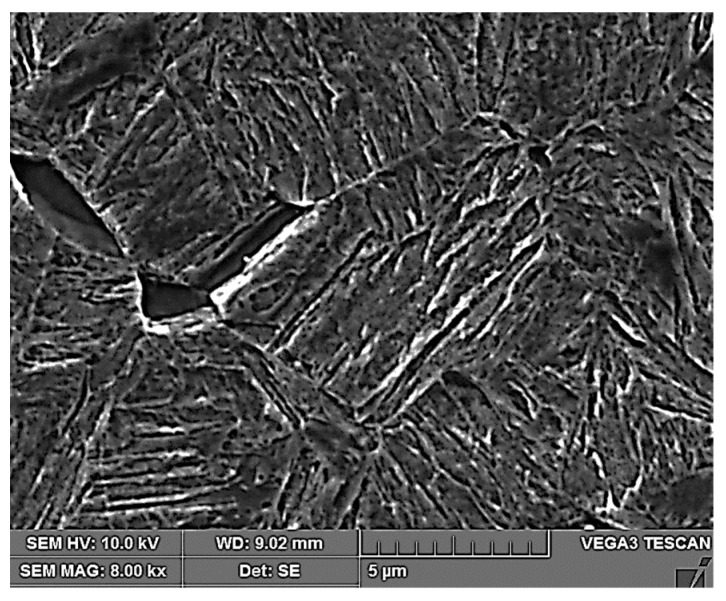
Structure after press hardening.

**Figure 19 materials-16-04340-f019:**
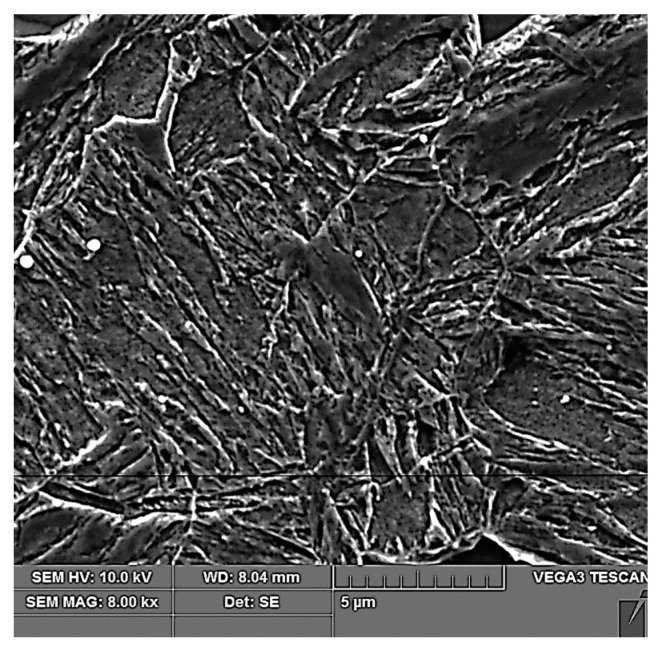
Tailored 10 + 5 cooled in the tool.

**Figure 20 materials-16-04340-f020:**
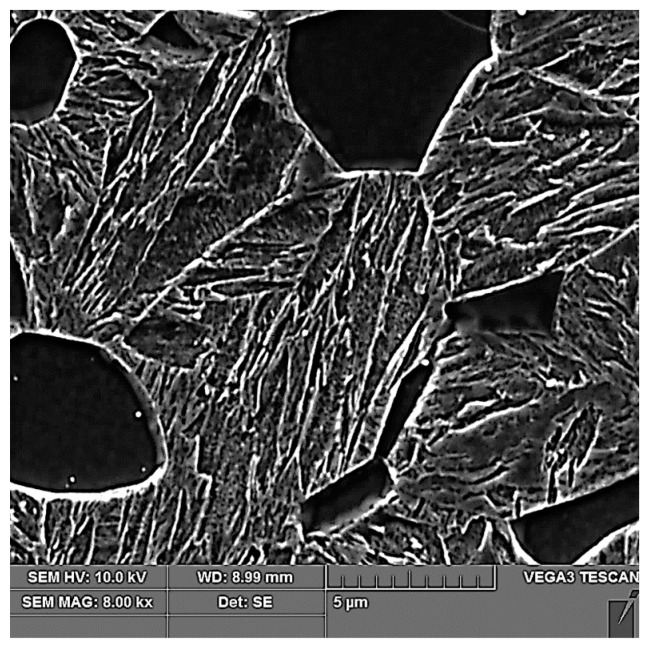
Tailored 10 + 5 cooled in the air.

**Figure 21 materials-16-04340-f021:**
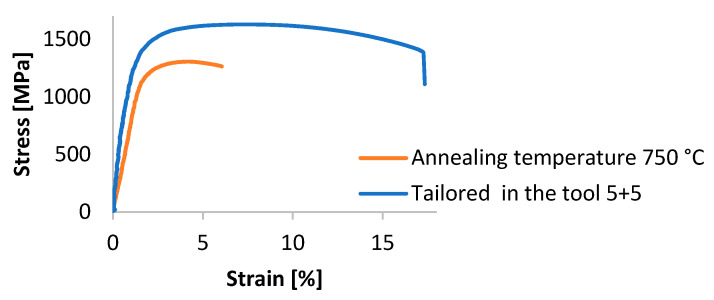
Progress of the tensile test.

**Figure 22 materials-16-04340-f022:**
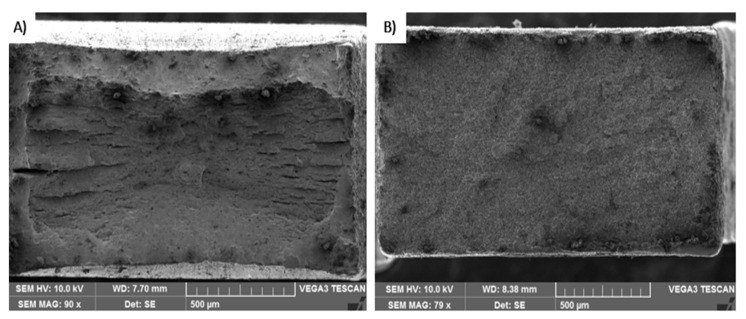
(**A**) Fracture surface tailored in the tool 5 + 5; (**B**) fracture surface annealing temperature 750 °C.

**Figure 23 materials-16-04340-f023:**
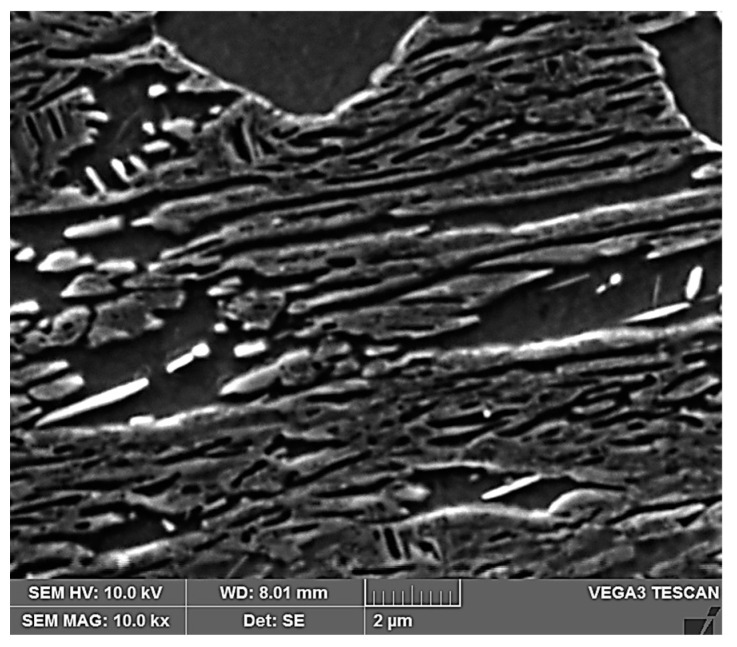
Structure after induction annealing.

**Table 1 materials-16-04340-t001:** Chemical composition of the material 2C5Mn3Al (wt%) determined by a Bruke Q4 Tasman optical spectrometer.

Chem. Composition	C	Si	Mn	P	S	Cr	Ti	Ni	Al
wt.%	0.2	0.58	5.02	0.0007	0.07	0.18	0.008	0.078	2.95

**Table 2 materials-16-04340-t002:** Temperatures of phase transformations for anisotropic cooling of steel.

Phase Transformation	Fs	A_c1_	Ps	A_c3_	Bs	M_s_
Temperature [°C]	1043	1071	685	683	444	286

**Table 3 materials-16-04340-t003:** Mechanical properties after press hardening.

Hardening Time in the Tool [s]	Annealing Temperature [°C]	Ultimate Strength [MPa]	Elongation [%]	Hardness HV10 [-]
1	-	1362 ± 9	17.7 ± 0.3	415 ± 6
5	-	1404 ± 10	16 ± 1	431 ± 3

**Table 4 materials-16-04340-t004:** Mechanical properties after intercritical annealing.

Hardening Time in the Tool [s]	Annealing Temperature [°C]	Ultimate Strength [MPa]	Elongation [%]	Hardness HV10 [-]
5	700	1261 ± 28	3.8 ± 1.2	273 ± 6
5	750	1323 ± 9	4.7 ± 1.5	335 ± 3
5	800	1425 ± 39	3.8 ± 2.8	377 ± 3

**Table 5 materials-16-04340-t005:** Temperatures and mechanical properties after processing by various tailoring methods.

Sample	Testing Area Sample	Temperature between the Step [°C]	Temperature at the End of the Process [°C]	Ultimate Strength [MPa]	Elongation [%]	Hardness HV10
5 + 5	In the air	695	390	1495 ± 15	12.4 ± 3.1	480 ± 5
	Transition	-	-	1585 ± 5	16 ± 1.1	489 ± 2
	In the tool	-	170	1635 ± 5	15.7 ± 0.9	506 ± 1
10 + 5	In the air	645	365	1440 ± 10	16.2 ± 1.8	477 ± 8
	Transition	-	-	1430 ± 20	14.7 ± 0.5	489 ± 5
	In the tool	-	118	1640 ± 0	14.7 ± 0.4	528 ± 9
30 + 5	In the air	520	275	1575 ± 5	18.3 ± 0.1	483 ± 3
	Transition	-	-	1555 ± 30	14.3 ± 2.3	471 ± 4
	In the tool	-	70	1555 ± 30	14.3 ± 0.8	465 ± 7

**Table 6 materials-16-04340-t006:** Mechanical properties of the material after induction annealing.

Ultimate Strength [MPa]	Elongation [%]	Hardness HV10 [-]
1219 ± 24.1	14.3 ± 1.5	396 ± 3

## Data Availability

The raw data are not publicly available due to ongoing research.

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
