# Peer review of "Effect of Forming and Heat Treatment Parameters on the Mechanical Properties of Medium Manganese Steel with 5% Mn"

_materials, 2023, doi:10.3390/ma16124340_

Round 1

Reviewer 1 Report

Effect of various different thermal treatments to a medium manganese steel with 5% Mn on its mechanical properties was presented. Although it seems obvious that the authors did a lot of experimental work and got some meaningful results such as good combination of strength and elongation of the steel after one of their many heat treatments, it was extremely difficult to read and understand what they claim out of the results because of poor and careless writing. It is highly skeptical if the manuscript could be improved to be readable by the normal revision process. It does not seem to be a problem of English writing only but together with a serious problem of writing itself.

Also, optical micrographs have too low visibility and are not actually providing any meaningful information. All the micrographs including OM and SEM images should be presented in close relation to what the authors are claiming in their main texts.

In summary, it seems necessary for this manuscript to be considered for publication that the writing should be improved to the level that the manuscript is readable. In that context, it is difficult to recommend publication of this manuscript. Some more detailed comments are as follows,

In the abstract, the part in lines 9-14 is not appropriate for abstract. Only key achievement of the study should be concisely presented in the abstract part. Also, abstract is for the readers who have not gone over the entire manuscript, therefore those terms like 'B columns' and 'omega profiles' that should not be familiar to those who have not gone over the manuscript should not be used.

The manuscript is full of sentences that is difficult to understand. Examples are as follows,

Line 38-39: The material must also allow for the necessary surface treatments, eliminating the use of certain alloying elements. [12]

Line 78-79: the deposition of manganese at different temperatures during heating of the material to IA temperature

Line 146: removal rate

Line 173-174: intergranular annealing

Line 182: figure caption of Figure 5 is not correct.

Line 188: omega pro-film

Line 183-200: this part is difficult to understand.

Line 207: CCT di-gram

Line 257: omega porphyries

Line 294: proetectoid

Line 402: is done by pressing to ensure high standards in order to protect health

Line 412: needles were visible sheets of residual austenite

As for the Table 3, what is the reason for the significant decrease in elongation after annealing?

Not only English language, but also writing itself should be improved significantly. It is highly recommended for the authors to consult to a professional writing specialist. Also, it is kindly recommended for the authors to go over the manuscript very carefully several times before they submit the manuscript.

Author Response

Comments and Suggestions for Authors:

Thank you for your comments and feedback. I have taken them into consideration and have reworked the article according to the revision. 

Effect of various different thermal treatments to a medium manganese steel with 5% Mn on its mechanical properties was presented. Although it seems obvious that the authors did a lot of experimental work and got some meaningful results such as good combination of strength and elongation of the steel after one of their many heat treatments, it was extremely difficult to read and understand what they claim out of the results because of poor and careless writing. It is highly skeptical if the manuscript could be improved to be readable by the normal revision process. It does not seem to be a problem of English writing only but together with a serious problem of writing itself.

Thank you for your feedback, I have gone through the text several times and changed most of the text in the Results and Discussion sections

Also, optical micrographs have too low visibility and are not actually providing any meaningful information. All the micrographs including OM and SEM images should be presented in close relation to what the authors are claiming in their main texts.

I have changed the figures and added captions to them for a better overview.

In summary, it seems necessary for this manuscript to be considered for publication that the writing should be improved to the level that the manuscript is readable. In that context, it is difficult to recommend publication of this manuscript. Some more detailed comments are as follows,

In the abstract, the part in lines 9-14 is not appropriate for abstract. Only key achievement of the study should be concisely presented in the abstract part. Also, abstract is for the readers who have not gone over the entire manuscript, therefore those terms like 'B columns' and 'omega profiles' that should not be familiar to those who have not gone over the manuscript should not be used.

I have rewritten the abstract and replaced terms like B column with other meanings.

The manuscript is full of sentences that is difficult to understand. Examples are as follows,

Line 38-39: The material must also allow for the necessary surface treatments, eliminating the use of certain alloying elements. [12]

Line 78-79: the deposition of manganese at different temperatures during heating of the material to IA temperature

Line 146: removal rate

Line 173-174: intergranular annealing

Line 182: figure caption of Figure 5 is not correct.

Line 188: omega pro-film

Line 183-200: this part is difficult to understand.

Line 207: CCT di-gram

Line 257: omega porphyries

Line 294: proetectoid

Line 402: is done by pressing to ensure high standards in order to protect health

Line 412: needles were visible sheets of residual austenite

 Thank you for your suggestions, I have replaced all of them with the correct terms.

As for the Table 3, what is the reason for the significant decrease in elongation after annealing?

I have developed the issue more in the discussion section

Comments on the Quality of English Language

Not only English language, but also writing itself should be improved significantly. It is highly recommended for the authors to consult to a professional writing specialist. Also, it is kindly recommended for the authors to go over the manuscript very carefully several times before they submit the manuscript.

Reviewer 2 Report

The authors provide an interesting research about the efect of different manufacturing parameters on the mechanical properties of a medium manganese steel. The topic deserves investigations and the findings have potential. However, the current version is quite far from being a document suitable for publication in a reputed journal:

1) First of all, the text is full of typos. Not necessary going here one by one, but readers should do so. I´ve counted dozens all along the text (e.g., "comsbination" in the abstract, decimals with commas or points indistinctly, acronyms presented without explanation, Tables and figures not mentioned in the text, etc)

2) The quality of the figures is generally poor, and some of them are well referred (Figs 1 and 2) but I guess they need further permission for publication. Analogously, in Figure 3, the different parameters should appear much more clearly. Figure 5 caption is, again, mistaken. Figure 6 is again of poor quality and the different trayectories should be identified.

3) The experimental programme may be followed qualitatively, but it is hard to follow quantitatively. Authors should clearly present before hand the different omega profiles to be tested, with their different conditions, in a new table. How many profiles are analyzed in total?

4) Then, they should explain the different measurements performed on each profile. How many tensile or hardness measurements? What locations? this info should appear clearly and sistematically.

5) The images showing the microstructures should identify the different microstructures.

6) Table 3 and Table 4 are not homogeneous, and they should be. Why elongation appears differently? What is A5mm? Why HV10 has standard deviation in Table 3 and not in Table 4?

7) More discussion is required about the low ductilities observed in the annealed profiles gathered in Table 3.

Please, dedicate time to this document. The research has clear potential, but the document is really poor.

English can be followed, but there are lots of typos, and some review is required concerning the grammar.

Author Response

Comments and Suggestions for Authors

The authors provide an interesting research about the efect of different manufacturing parameters on the mechanical properties of a medium manganese steel. The topic deserves investigations and the findings have potential. However, the current version is quite far from being a document suitable for publication in a reputed journal:

Thank you for your comments and feedback. I have taken them into consideration and have reworked the article according to the revision. 

1) First of all, the text is full of typos. Not necessary going here one by one, but readers should do so. I´ve counted dozens all along the text (e.g., "comsbination" in the abstract, decimals with commas or points indistinctly, acronyms presented without explanation, Tables and figures not mentioned in the text, etc)

Thank you for your feedback, I have gone through the text and removed the flaws.

2) The quality of the figures is generally poor, and some of them are well referred (Figs 1 and 2) but I guess they need further permission for publication. Analogously, in Figure 3, the different parameters should appear much more clearly. Figure 5 caption is, again, mistaken. Figure 6 is again of poor quality and the different trayectories should be identified.

Figures have been changed and captions corrected

3) The experimental programme may be followed qualitatively, but it is hard to follow quantitatively. Authors should clearly present before hand the different omega profiles to be tested, with their different conditions, in a new table. How many profiles are analyzed in total?

Incorporated into the text of the section Evaluation methods applied:

In total, 1 omega profile with a tool holding time of 1 s and 4 omega profiles with a tool holding time of 5 s were produced. Three of these profiles were then used for other parts of the experiment (intercritical annealing and local induction heating).

A metallographic sample, in the transverse direction, was prepared from each omega profile, hardness measurements and tensile tests were performed. For tensile test we used two samples.

Furthermore, three omega profiles were produced for the "Production of an omega profile with locally different properties" part, where the endurance parameters were selected.

One metallographic cut (metallography and hardness measurements) was produced from these omega profiles and two tensile tests were produced for each area under investigation (in the air, Transition, In the tool).

We have divided the results table so that under each chapter it is clear what analyses were performed.

4) Then, they should explain the different measurements performed on each profile. How many tensile or hardness measurements? What locations? this info should appear clearly and sistematically.

The direction of the samples for tensile tests can be seen in Fig. 8. The number of samples and the number of measurements have been incorporated into the text.

5) The images showing the microstructures should identify the different microstructures.

I have changed the figures and added captions to them for a better overview.

6) Table 3 and Table 4 are not homogeneous, and they should be. Why elongation appears differently? What is A5mm? Why HV10 has standard deviation in Table 3 and not in Table 4?

Tabs 3 and 4 are the same now, I forgot to add the standard deviation there. I have incorporated the different extensions into the text in the discussion. A5mm was a marking from our machine, it is now corrected.

7) More discussion is required about the low ductilities observed in the annealed profiles gathered in Table 3.

. The problem was described more thoroughly in the discussion.

Please, dedicate time to this document. The research has clear potential, but the document is really poor.

Comments on the Quality of English Language

English can be followed, but there are lots of typos, and some review is required concerning the grammar.

Reviewer 3 Report

The manuscript describes the “Effect of forming and heat treatment parameters on the mechanical properties of medium manganese steel with 5% Mn”, which is suitable for Materials. However, the reviewer would like to make the following comments;

 Abstract

· It has to be improved. Mentioned the values, improvement percentage, and significant findings.

 Introduction

·The quality of Figures 1 and 2 is low.

·Mentioned the results of the cited references with values and compared the results.

· The problem statement of the research needs to be clarified.

Experimental procedure

·Present the sample preparation as a schematic diagram in a Figure.

·The quality of Figure 3 is low.

·Present the samples' location for Microstructure observation and tensile test by a schematic diagram.

· Mention the standard code for tensile and hardness tests used in the study.

  Results and Discussion

·Menion the name of different phases (ferrite, martensite, residual austenite, globular austenite, proetectoid ferrite) on microstructures figures i.e. Figures 10-16.

·  Present the stress-strain curves for samples mentioned in Tables 3 and 4.

·  Present the macro photo and fracture surfaces of samples after the tensile test.

Author Response

Comments and Suggestions for Authors

The manuscript describes the “Effect of forming and heat treatment parameters on the mechanical properties of medium manganese steel with 5% Mn”, which is suitable for Materials. However, the reviewer would like to make the following comments;

Thank you for your comments and feedback. I have taken them into consideration and have reworked the article according to the revision. 

 Abstract

  • It has to be improved. Mentioned the values, improvement percentage, and significant findings.

 I added the values from the results to the abastract and modified it.

Introduction

  • The quality of Figures 1 and 2 is low.

Figures have been changed

  • Mentioned the results of the cited references with values and compared the results.

The cited references and their values were compared in the discussion.

  • The problem statement of the research needs to be clarified.

This has been incorporated and edited into the text at the end of the introduction section.

Experimental procedure

  • Present the sample preparation as a schematic diagram in a Figure.

A schematic figure with captions has been added.

  • The quality of Figure 3 is low.

The figure has been redrawn

  • Present the samples' location for Microstructure observation and tensile test by a schematic diagram.

This information with figure 8. and has been incorporated and edited into the text.

  • Mention the standard code for tensile and hardness tests used in the study.

This information has been incorporated and edited into the text in the section Evaluation methods applied

 Results and Discussion

  • Menion the name of different phases (ferrite, martensite, residual austenite, globular austenite, proetectoid ferrite) on microstructures figures i.e. Figures 10-16.

I have changed the figures and added captions to them for a better overview.

  • Present the stress-strain curves for samples mentioned in Tables 3 and 4.

stress-strain curves is presented on page 15

  • Present the macro photo and fracture surfaces of samples after the tensile test.

Photos are described and incorporated into the text on page 15

Round 2

Reviewer 2 Report

The paper has been improved and provides now interesting findings.

I am still concerned about the use of figures from other sources, so the editiorial should pay attention to this.

The English is good enough. Last minor revision would be appreciated.

Reviewer 3 Report

The authors responded to all reviewers' comments in the revised version. Therefore, the revised manuscript can be considered for publication.